# Functional and Bioactive Properties of Protein Extracts Generated from *Spirulina platensis* and *Isochrysis galbana T-Iso*

**Stephen Bleakley** [1,2] **and Maria Hayes** [1,*]

1   Food Biosciences, Teagasc Food Research Centre, Ashtown, 15 Dublin, Ireland;
    Stephen.Bleakley12@gmail.com

2   School of Biological Sciences, College of Sciences and Health and Environment,
    Sustainability and Health Institute, DIT Kevin Street, D08 NF82 Dublin, Ireland

*   Correspondence: Maria.Hayes@teagasc.ie; Tel.: +353−1−8059957

**Featured Application: The heart health-beneficial activities of algal proteins extracted using a protein extraction method involving aqueous sonication, precipitation and dialysis were identified. In addition, proteins were found to have techno-functional attributes that could potentially be used in the development of meat substitute products or alkaline beverages, based on the oil-holding capacities and solubility of the proteins.**

**Abstract:** There is growing consumer demand for food products derived from microalgae, driven largely by the perceived health benefits associated with them. The functional and bioactive potential of proteins isolated from two microalgae—*Spirulina* sp. and *Isochrysis galbana T-Iso*—were determined. The results obtained show the potential of microalgal protein extracts for use in the beverage industry, based on solubility values and other functional characteristics, including water and oil holding capacities, foaming, emulsifying activities and stabilities, water activities, solubility and pH. The solubility of algal proteins was pH-dependent, and they were largely insoluble at pH values between 2 and 11. However, the proteins were increasingly soluble at a pH of 12, and they have potential use in formulating foods with higher viscosities or gels, where they could act as fillers to strengthen networks. Compared with whey and flaxseed proteins, the *Spirulina* sp. protein extract had a superior oil-holding capacity (OHC). The OHC is important in developing texture in food products such as meats. Overall, better foam stability was observed for both *Spirulina* sp. and *Isochrysis* sp. soluble protein extracts, compared with flaxseed protein at pH values from 2 to 10 over a period of 120 min. The foam capacity and stability increase the physical properties of foods. However, the emulsion activity and stability values for soluble algal protein extracts were less than the values observed for flaxseed and whey proteins. Algal proteins would not be suitable for use in creaming and food processing involving flocculation. In addition, algal protein extracts inhibited Angiotensin-converting enzyme−1 (ACE−1) and renin, and they have potential for use in functional food ingredient applications to maintain heart health and also to act as meat substitutes.

**Keywords:** *Spirulina* sp.; *Isochrysis galbana T-Iso*; angiotensin-I-converting enzyme (ACE−1) inhibition; renin inhibition; protein; solubility; emulsifying; foaming; water activity

## 1. Introduction

There are over 30,000 microalgae species; however, fewer than ten species are commercially produced [1, 2], and only two species are recognized for sale within the European Union with European Food Safety Authority (EFSA) approval. Select microalgae contain protein levels based on a dry matter basis, similar to some plant and animal sources of protein. The reported protein content of *Spirulina platensis* varies from 65% to

77% of the dry matter, a value greater than the protein content of soy flour (37%), chicken (24%), fish (24%), beef (22%) and skimmed milk (36%) [3]. *Isochrysis* sp. reportedly contains 30.5% protein on a dry weight basis [4]. Although microalgae were used throughout history as a source of food [5], the industrial production of microalgal protein is still a developing field of research due to technical difficulties regarding cultivation, cost of cultivation and processing and limited knowledge regarding their functional, nutritional and bioactive attributes.

Several research groups have looked at extracting proteins from microalgae, including *Spirulina* sp. previously. Ultrasound followed by sugaring out with ionic flotation [6] and ultrasound alone were applied to *Spirulina* sp. previously to disrupt the cell wall [7]. Previously, Safi et al. determined the proportion of proteins released from the microalgae *Porphyridium cruentus*, *Arthrospira platensis*, *Chlorella vulgaris*, *Nannochloropsis oculata* and *Haematococcus pluvialis*, following both mechanical and chemical treatments with a high-pressure disruptor and 2N NaOH [8]. The proportion of essential and non-essential amino acids in the extracts was higher in the alkaline and high-pressure treated samples compared with the untreated algal biomass [8]. A study by Garcia-Vaquero et al. assessed the functional properties of protein extracts generated from the brown alga *Himanthalia elongata* (Linnaeus) S. F. Gray [9]. This study suggested that protein extracted from this alga could be suitable for use in the formulation of a wide variety of food products, including sausages, breads, cakes, soups and salads [9]. What protein extraction method is considered the best method depends on the alga treated and the structure of the algal cell wall. Cost is also a factor that should be considered. The type of alga treated, seasonal variations in the quantities of proteins in the starting materials and method all play a part in determining the yield of protein obtainable from a microalga. In addition, the health properties of *Spirulina* sp. proteins were looked at previously [10]. Protein extracts from microalgae, including *Dunaliella* sp., at an industrial scale can produce more protein compared with terrestrial plant harvests (about 100 times greater yield).

The aim of this work was to generate protein extracts from the algae *Spirulina platensis* and *Isochrysis galbana T-Iso* using an aqueous sonication method, followed by precipitation with ammonium sulphate and dialysis procedure, as described previously by Galland-Irmouli et al. [11]. The functional and bioactive attributes of the extracted proteins were determined and compared to protein sources including whey protein isolate and flaxseed protein, which are used widely in the food industry as techno-functional ingredients.

## 2. Materials and Methods

### 2.1. Materials and Algae Used in This Study

Microalgae were kindly cultivated and supplied by the University of Almeria in Andalucía, Spain. *Isochrysis galbana T-Iso* was grown in tubular reactors using saltwater and fertilizers as a cultivation medium. *Spirulina platensis* was cultivated in an open raceway reactor using fertilizers as a growth medium and with pH control by means of the addition of sodium bicarbonate. The samples were received in freeze-dried form and subsequently stored at −80 °C until further use. BioZate®, a whey protein isolate (BioZate, Davisco Foods, USA), and defatted golden flaxseed meal (Glanbia Nutritionals, USA) were used as dairy and cereal protein comparisons in the assessment of the functional properties of the microalgal proteins. Sigma Aldrich (Sigma Aldrich, Dublin, Ireland) supplied the ammonium sulphate. Angiotensin−1-converting enzyme (ACE−1; EC 3.4.15.1) inhibition was determined using the ACE−1 inhibition colorimetric assay kit supplied by Dojindo Laboratories (Dojindo Laboratories, Kumamoto, Japan). Captopril© was supplied by Sigma Aldrich (Steinheim, Germany). Renin inhibition was determined using a renin activity fluorometric assay kit from BioVision Inc. (Cambridge bioscience, Cambridge, UK). The specific renin inhibitor Z-Arg-Arg-Pro-Phe-His-Sta-Ile-His-Lys-(Boc)-OMe, which was used as a positive control as detailed in previous studies [12,13], was supplied by Sigma Aldrich (Steinheim, Germany).

### 2.2. Protein Extraction and Determination

Crude protein extracts were generated from microalgae using the method previously described by Galland-Irmouli et al., which will be referred to as the traditional protein extraction method [11]. Briefly, microalgae were added independently to Milli-Q ultrapure water at a concentration of 2% (*w/v*) and sonicated using a Bran sonic® 3510EMT (Branson Ultrasonic SA, Switzerland)) at 42 kHz for 1 h. The treated biomass was stirred overnight at 4 °C and centrifuged (Sorvall LYNX 6000 super speed centrifuge, Thermo Scientific, UK) at 10,000× *g* for 1 h at 4 °C. The supernatant was subsequently collected and stored at 4 °C, while the pellet was resuspended in ultrapure water at a concentration of 4% (*w/v*). The resuspended biomass was sonicated for 1 h, stirred overnight at 4°C and centrifuged at 10,000× *g* for 1 h at 4°C. The supernatants were pooled and saturated with 80% ammonium sulphate (Sigma Aldrich, Dublin, Ireland). This was followed by stirring for 1 h at 4 °C and centrifugation at 17,000× *g* for 1 h at 4 °C. The pellet was resuspended and dialyzed overnight at 4 °C using Thermo scientific snakeskin™ 3.5 kDa molecular weight cut-off (MWCO) tubing (Fischer Scientific, New Hampshire, USA). The dialyzed protein extracts were freeze-dried using an industrial-scale freeze drier, namely the FD 80 model (Cuddon Engineering, Marlborough, New Zealand), and subsequently stored at −20 °C until further use.

The generated protein extracts underwent initial proximate quality analysis to determine the protein, ash and lipid content. The yield of extracted protein relative to the dry biomass of the freeze-dried extracts was determined using a LECO FP628 protein analyzer (LECO Corp., MI, USA) based on the Dumas method and according to AOAC method 992.15, as referenced previously [14]. A conversion factor of 4.4 was used to compute the protein content of the extracts in the microalgal protein extracts [15]. The yield of extracted protein relative to the dry biomass was calculated as g protein in the crude protein extract / g alga dry weight (DW) biomass.

### 2.3. Total Amino Acid (TAA) and Free Amino Acids (FAAs) Composition

The total amino acid composition (free and bound amino acids) and free amino acids (free) of the protein extracts generated from *Spirulina* sp. and *Isochrysis* sp. were determined. To determine the TAA, the protein extracts were hydrolyzed using 6 M HCL at 110 °C for 23 h, as described previously by Hill et al. [16]. The samples were then deproteinized by mixing equal volumes of 24% (*w/v*) tri-chloroacetic acid and the sample. These were allowed to stand for 10 min at room temperature before centrifugation at 14,400 x G for 10 min. The supernatants were removed and diluted with a 0.2 M sodium citrate buffer at a pH of 2.2 to give approximately 250 nmol of each amino acid residue, based on the known molecular masses of amino acids. The samples were then diluted 1 in 2 with the internal standard norleucine to give a final concentration of 125 nmol/mL. Amino acids were quantified using a Jeol JLC−500/V amino acid analyzer (Jeol (UK) Ltd., Garden city, Herts, UK) fitted with a Jeol Na+ high-performance cation exchange column. The free amino acid analysis was carried out without sample hydrolysis and by diluting the samples 1 in 2 with an internal standard (norleucine) to give a final concentration of 125 nmol/L. FAAs were quantified using the Jeol JLC- 500/V amino acid analyzer (Jeol (UK) Ltd., Garden city, Herts, UK).

### 2.4. Protein Size Composition and Polyacrylamide Gel Electrophoresis

The protein size composition was measured by preparing 1% algal protein extracts (*w/v*) with ultrapure water adjusted to a pH of 12 using 1 M NaOH to increase the solubility of the protein. The samples were shaken for 45 min and centrifuged at 4000× *g* for 30 min. The insoluble protein pellet was stored at −20 °C. The Amicon ultra−15 centrifugal filter units (Merck Millipore, Cork, Ireland) were washed with ultrapure water. Algal protein suspensions were first added to these units with a 3-kDa molecular weight cut-off (MWCO) filter and centrifuged at 5000× *g* for 1 h, as per the manufacturer's instructions.

The permeate was stored at −20 °C, while the retentate was made up to 12 mL and added to the filter units with a 10 kDa MWCO and centrifuged at 5000× $g$ for 30 min. This process was repeated using 30, 50 and 100 kDa MWCO filters. The permeates were stored at −20 °C. The samples were freeze-dried using an industrial-scale freeze drier, namely the FD 80 model (Cuddon Engineering, Marlborough, New Zealand), then weighed, and the percentage values were calculated. Tris-tricine SDS-PAGE was performed using a Mini-PRO-TEAN® electrophoresis unit (Bio-Rad laboratories, USA) using Mini-PROTEAN® 10–20% Tris-tricine precast gels (Bio-Rad laboratories, USA). Protein separation was performed according to the manufacturer's instructions. Briefly, the protein extracts dissolved in ddH$_2$O (20 mg/mL) were diluted 1:1 with a loading buffer containing 200 mM Tris-HCL pH 6.8, 2% SDS, 40% glycerol, 0.04% Coomassie Blue G−250 and 350 mM DTT. The samples were heated at 95 °C for 10 min and loaded (10 uL of each sample in each well) in the pre-cast gels in the electrophoresis unit in the presence of a running buffer containing 100 mM Tris-base, 100 mM tricine and 0.1% SDS. The running conditions were 30–35 mA for 3 h and 15–20 mA for 2 h. The pre-stained precision plus protein TM Dual Xtra molecular mass marker (250–2 kDa, Bio-Rad, Dublin, Ireland) was used. All images were analyzed using Quantity One® software version 4.5.2 (Bio-Rad laboratories, USA).

### 2.5. Determination of the pH of Protein Extracts

The freeze-dried algal protein extracts were resuspended in ultrapure water at a concentration of 1% (*w/v*), and the pH was measured using a pH meter (Orion, model 420 A – Thermo Orion, Cambridge shire, UK).

### 2.6. Water Activity (a$_w$)

The water activity (a$_w$) value of the freeze-dried algal protein extracts was measured using an Aqua Lab water activity system (Pullman, Washington, USA). The water activity was measured at 22.4 ± 1.6 °C.

### 2.7. Water-Holding Capacity (WHC) and Oil-Holding Capacity (OHC) of Protein Extracts

The water-holding capacity (WHC) and oil-holding capacity (OHC) of the generated protein extracts were assessed according to the previously published method of Bencini [17] with slight modifications. The algal protein extracts were mixed with water or vegetable oil at a concentration of 1% (*w/v*) using a vortex mixer (Henry Troemner, USA) for 1 min. The mixed protein extracts were allowed to stand for 30 min and then centrifuged at 2,200× g for 30 min. The supernatants were decanted, and the tubes containing the pellet were weighed. The WHC and OHC were calculated as grams of water or oil held by 1 g of algae protein extract using the following formula:

$$\text{WHC/OHC} = \text{g H}_2\text{O or oil/g protein extract} = W_2 - W_1/W_0 \text{ X}100$$

where $W_0$ is the weight of the dry sample (g), $W_1$ is the weight of the tube plus the dry sample (g) and $W_2$ is the weight of the tube plus the pellet following removal of the supernatant (g).

### 2.8. Solubility

The solubility of the algal protein extracts was determined based on the method by Beuchat et al. [18] with slight modifications. The algal protein extracts were prepared in ultrapure water at a concentration of 1% (*w/v*), and the pH was adjusted to between 2 and 12 using 1 M NaOH and 1 M HCl. The suspended samples were mixed at room temperature for 45 min using a Multi-Reax vibrating shaker (Heidolph, Germany), and centrifuged at 4000× $g$ for 30 min at 4 °C. The amount of soluble protein was determined in the supernatant using the Quanti-Pro BCA Assay Kit (Sigma, St. Louis, USA) as per the manufacturer's instructions. The percentage of solubility of the protein extract at each pH

point was calculated based on the total protein content at full dispersion using the following formula:

$$S\ (\%) = (C_{sup} / C_{tot}) * 100\%$$

where S is the solubility, C is the concentration of protein, $C_{sup}$ is the concentration of protein in the supernatant measured using the BCA assay and $C_{tot}$ is the concentration of protein in the total fraction.

### 2.9. Foaming Capacity and Stability

The foaming capacity (FC) of the algal protein extracts was examined using the method outlined by Poole et al. with slight modifications [19]. The algal protein extracts were prepared at a concentration of 1.5% (*w/v*) in ultrapure water. The pH values of the samples were adjusted to2, 4, 6, 8 and 10 using 1 M NaOH and 1 M HCl. The protein suspensions were homogenized using a T25 Ultra-Turrax homogenizer (IKA®, Germany) for 1 min at 10,000 RPM. The volumes of the protein suspensions before homogenization and foam generated after homogenization were measured using a graduated cylinder. $F_C$ was calculated as a percentage of the initial protein suspension volume using the following formula:

$$F_C = V_F / V_0 \times 100$$

where $F_c$ is the foaming capacity, $V_F$ is the volume of foam after homogenization and $V_0$ is the initial volume of the protein suspension before homogenization. Foaming stability was expressed as a percentage of the initial foam volume and determined by measuring the volume of foam at 15, 30, 60, 90 and 120 min after homogenization.

### 2.10. Emulsifying Activity and Stability (EAI and ESI) of Protein Extracts

The emulsion activity index (EAI) was carried out according to the turbidimetric method outlined by Pearce and Kinsella [20] with slight modifications. A 1% protein solution was made by adding a sample amount corresponding to 120 mg protein to 12 mL ddH$_2$O. Four milliliters of sunflower oil, rapeseed oil, olive oil or groundnut oil was then added separately to the protein solution and homogenized using a T25 Ultra-Turrax homogenizer (IKA®, Germany) for 1 min at 14,000 RPM. A 50 μL aliquot was taken from the bottom of the tube and added to 5 mL of 0.1% SDS at 0 and 10 min after homogenization. The absorbance of the diluted solution was measured at 500 nm, and the EAI and ESI were calculated as follows:

$$EAI\ (m^2/g) = 2 \times T \times A_0 / \theta \times c$$

where T is the turbidity (2.303), $A_0$ is the absorbance measured at 500 nm 0 min after homogenization, $\theta$ is the oil volumetric fraction of the homogenized solution (0.25) and c is the protein concentration (0.1 g/mL):

$$ESI\ (min) = A_0 \times \Delta_t / \Delta T$$

where $\Delta_t$ is the change in time (10 min) and $\Delta T$ is the change in turbidity ($A_0 - A_{10}$).

### 2.11. ACE-I and Renin Inhibitory Activity of Extracted Soluble Proteins

An ACE-I (angiotensin-converting enzyme I) inhibition bioassay of the microalgal proteins was carried out according to the manufacturer's instructions (ACE Kit—WST, Dojindo Laboratories, Kumamoto, Japan). Briefly, 20 μL of each sample of aqueous solution at a concentration of 1 mg/mL was added to 20 μL of substrate and 20 μL of an enzyme working solution in triplicate. Captopril© was used as a positive control at a concentration of 0.05 μM, as recommended by the manufacturer. The samples were incubated at 37 °C for 1 h. A 200 μL indicator working solution was then added to each well, and subsequent incubation at room temperature was carried out for 10 min. The absorbance

at 450 nm was read using a FLUOstarOmega microplate reader (BMG LABTECH GmbH, Offenburg, Germany). The percentage of inhibition was calculated using the following equation:

$$\text{ACE-I Inhibition (\%)} = ((A_0 - A_I)/A_0) * 100\%$$

where $A_0$ is the substrate absorbance at 450 nm in the presence of ACE-I and absence of the inhibitor and $A_I$ is the substrate absorbance at 450 nm in the presence of ACE-I and the inhibitor or Captopril© (positive control).

Protein isolates from the microalgae were tested in vitro for renin inhibition activity. The renin inhibition screening assay (Cambridge Biosciences, Cambridge, UK) was carried out as per the manufacturer's instructions. Briefly, 10 µL of each sample of the inhibitor or renin-positive control Z-Arg-Arg-Pro-Phe-His-Sta-Ile-His-Lys-(Boc)-OMe at a concentration of 1 mg/mL dimethyl sulfoxide (DMSO) was added independently to the 20 µL renin substrate, 150 µL assay buffer and 10 µL renin in triplicate. The samples were incubated at 37 °C for 15 min and read with excitation wavelengths of 340 nm and emission wavelengths of 500 nm. The fluorescence was read using a FLUOstar Omega microplate reader (BMG LABTECH GmbH, Offenburg, Germany). The percentage of inhibition was calculated using the following equation:

$$\text{Renin Inhibition (\%)} = ((A_0 - A_I)/A_0) * 100\%$$

where $A_0$ is the substrate florescence in the presence of renin and absence of the inhibitor and $A_I$ is the substrate florescence in the presence of renin and the inhibitor or control (Z-Arg-Arg-Pro-Phe-His-Sta-Ile-His-Lys-(Boc)-OMe, (Sigma-Aldrich, Steinheim, Germany)).

### 2.12. Statistical Analysis

All measurements were carried out in triplicate (n = 3). Statistical analysis was performed using Excel 2010. The differences in solubility and foaming capacity at different pH values were analyzed using one-way ANOVA and a post-hoc Tukey's HSD test. In all cases, the criterion for statistical significance was $p < 0.05$.

## 3. Results and Discussion

### 3.1. Protein Yield and Composition

The protein contents of the generated extracts isolated from *Spirulina platensis* and *Isochrysis galbana T-Iso* were calculated, and the results are shown in Table 1. The extracts contained 85.50 ± 4.90% and 71.90 ± 8.60% protein, respectively. The *Spirulina sp.* and *Isochrysis sp.* extracts contained 3.27-± 4.90% and 2.87 ± 1.27% lipid and consisted of 2.10 ± 0.40% and 1.70 ± 1.10% ash, respectively. The quantity of extractable protein was 17.15 ± 1.5 g protein / 100 g from the *Spirulina* sp. biomass and 23.18 ± 10.60 g protein / 100 g biomass from the *Isochrysis* sp. biomass, respectively, using the traditional extraction method. These results are lower than the protein yields described in previous studies, which reported the level of protein obtainable from *Spirulina platensis* to be 55–77% of the dry weight of the alga. This may be due to the method used to cultivate *Spirulina* sp. and *Isochrysis* sp. for use in this work. It may also be due to a different nitrogen conversion factor being used in previous calculations, which resulted in a higher estimation of protein in these studies. It is well documented that protein measurement methods can yield very different results [21, 22, 23]. In addition, the use of sonication to burst the cell wall may have involved a probe instead of a sonication bath, as used in this work.

**Table 1.** Proximate analysis and total amino acid content of the microalgal protein extracts.

| | | *Spirulina* sp. | | *Isochrysis* sp. | |
|---|---|---|---|---|---|
| | | *Spirulina platensis* Whole Biomass (g/kg) | *Spirulina platensis* Protein Extract (g/kg) | *Isochrysis* T-Iso Whole Biomass (g/ kg) | *Isochrysis* T-Iso Protein Extract (g/kg) |
| **Essential amino acids** | Isoleucine | 3.53 | 4.61 | 2.75 | 0.3 |
| | Leucine | 5.59 | 7.58 | 5.39 | 0.45 |
| | Valine | 4.54 | 5.46 | 3.64 | 0.4 |
| | Phenylanine | 2.95 | 4.38 | 3.07 | 0.3 |
| | Tyrptophan | . | . | . | . |
| | Histidine | 1.48 | 1.82 | 2.53 | 0.59 |
| | Lysine | 2.78 | 2.64 | 2.8 | 0.27 |
| | Threonine | 3.38 | 3.58 | 3.13 | 0.31 |
| | Methionine | 1.84 | 1.9 | 1.95 | 0.17 |
| **Non-essential amino acids** | Alanine | 5.38 | 5 | 4.88 | 0.54 |
| | Glycine | 3.28 | 4.11 | 3.46 | 0.36 |
| | Proline | 2.35 | 3.38 | 3.17 | . |
| | Tyrosine | 2.41 | 2.75 | 0.98 | 0.38 |
| | Aspartic acid and Asparagine | 6.55 | 7.18 | 5.92 | 0.72 |
| | Glutamic acid and Glutamine | 8.99 | 7.65 | 6.43 | 0.73 |
| | Arginine | 4.09 | 4.31 | 3.61 | 0.38 |
| | Serine | 3.06 | 3.63 | 2.72 | 0.27 |
| | Cysteine | 1.56 | 0.69 | 1.55 | 0.48 |
| **Total amino acids (TAA)** | | 63.76 | 70.67 | 57.98 | 6.65 |
| **Essential amino acids (EAA)** | | 26.09 | 31.97 | 25.26 | 2.79 |
| **Non-essential amino acids (NEAA)** | | 47.15 | 38.7 | 32.72 | 3.86 |
| **Protein** | | | 85.5 ± 4.9% | | 71.9 ± 8.6% |
| **Ash** | | | 2.1 ± 0.4% | | 1.7 ± 1.1% |
| **Lipid** | | | 3.27 ± 4.9% | | 2.87 ± 1.27% |

*3.2. Amino Acid Composition*

The total amino acid compositions of the *Spirulina* sp. protein extract and *Isochrysis* sp. protein extract are shown in Table 1. The total amino acid contents of the whole *Spirulina* sp. and *Spirulina* sp. protein extracts were determined to be 314.71 g/kg DW and 380.99 g/kg DW, respectively, and the total amino acid contents of *Isochrysis* sp. and the protein extracts were determined to be 257.06 g/kg DW and 22. 20g/kg DW, respectively, with the essential amino acid composition of approximately 40.93% of the total amino acid composition for *Spirulina* sp. and 45.18% for the *Spirulina* protein extract. The essential amino acid composition for the *Isochrysis* sp. and *Isochrysis* sp. protein extracts were 43.55% and 42.03% of the total amino acid composition, respectively, and were comparable to other reports [24]. The level of the essential amino acids lysine and methionine contained in both the soluble protein extracts were 14.26 μg/mL and 9.08 μg/mL of lysine, respectively, for the *Spirulina* sp. soluble protein extract and *Isochrysis* sp. soluble protein extract and 0.913 μg/mL and 0.57 μg/mL of methionine, respectively. The *Spirulina* sp.

soluble protein extracts contained less methionine and lysine than previously reported by Liestianty et al. [25], who reported methionine and lysine contents of 14 mg g−1 (methionine) and 30 mg g−1 (lysine). The free amino acid contents of both microalga protein extracts are shown in Table 2.

**Table 2.** Free amino acid contents of the whole alga and protein extracts.

| | | *Spirulina* sp. | | *Isochrysis* sp. | |
|---|---|---|---|---|---|
| | | *Spirulina platensis* **Whole Biomass (g/kg)** | *Spirulina platensis* **Protein Extract (g/kg)** | *Isochrysis T-Iso* **Whole Biomass (g/kg)** | *Isochrysis T-Iso* **Protein Extract (g/kg)** |
| **Essential amino acids** | Isoleucine | 10.42 | 1.84 | 12.05 | 0.45 |
| | Leucine | 25.53 | 2.31 | 17.63 | 0.55 |
| | Valine | 15.2 | 1.96 | 15.86 | 1.05 |
| | Phenylalanine | 32 | 2.63 | 12.3 | 0.54 |
| | Tyrptophan | 6.73 | . | 15.79 | . |
| | Histidine | 20.44 | 1.53 | 7.12 | 0.69 |
| | Lysine | 8.98 | 0.13 | 4.54 | . |
| | Threonine | 7.16 | 0.94 | 16.91 | 0.64 |
| | Methionine | 4.92 | . | 2.51 | . |
| **Non-essential amino acids** | Alanine | 24.37 | 1.33 | 50.88 | 1.84 |
| | Glycine | 3.92 | 0.55 | 12.45 | 0.14 |
| | Proline | 1.14 | 0.22 | 18.32 | . |
| | Tyrosine | 29.55 | 2.81 | 12.01 | 0.45 |
| | Aspartic acid and Asparagine | 6.08 | 1.08 | 7.43 | 0.69 |
| | Glutamic acid and Glutamine | 54.79 | 2 | 11.01 | 0.89 |
| | Arginine | 15.12 | 1.79 | 20.98 | . |
| | Serine | 8.65 | 0.56 | 10.14 | 0.59 |
| | Cysteine | 17.18 | 2.51 | 6.19 | 1.16 |
| **Total free amino acids (FAA)** | | 292.18 | 24.19 | 254.12 | 9.68 |
| **EAA** | | 131.38 | 11.34 | 104.71 | 3.92 |
| **NEAA** | | 160.8 | 12.85 | 149.41 | 5.76 |

The protein content of the *Spirulina* sp. Extract, in terms of TAA and FAAs, was greater following extraction than that of *Isochrysis* sp., despite the fact that both whole microalgae have similar total and free amino acid profiles. Both microalgae contained a greater percentage of FAA (292.18 g/kg and 254.12 g/kg for the *Spirulina* sp. and *Isochrysis* sp., respectively) than TAAs (63.76 g/kg and 57.98 g/kg for *Spirulina* sp. and *Isochrysis* sp. respectively). The extraction process increased the TAA content of the *Spirulina* sp. protein extract, but the *Isochrysis* sp. protein extract contained less than 10% of the available TAA in this microalga. The *Isochrysis T-Iso* species does not contain a cell wall. The protein extraction process used was not suitable for this species, and this warrants further investigation. The yield of protein obtained from *Isochrysis* sp. in this study was similar to that achieved previously with pressurized liquid extraction using water, where approximately 18% of the protein was recovered [25]. Furthermore, there were inconsistencies observed between the determined protein content (% values) and the TAA content for the two extracts. Both contained >70% protein (i.e., >700 g/kg), but the total AA content was 7% (i.e., 70 g/kg). Therefore, for *Spirulina* sp., the discrepancy was a factor of 12, while for *Isochrysis* sp., it was a factor of 108. This may be due in part to the protein determination method used and discussed and the nitrogen conversion factor used.

### 3.3. Determination of pH and Water Activity of Protein Extracts

The water activity ($a_w$) values of the protein extracts generated from *Spirulina* sp. and *Isochrysis* sp. were determined to be 0.431 ± 0.01 and 0.389 ± 0.03, respectively, when measured at 22.4 °C ± 0.78 °C. The water activity values observed for the protein extracts would not permit the growth of microbes (yeasts, molds, spoilage bacteria or fungi), as they are below 0.45. The protein extracts can therefore be considered stable and should not support microbial growth. The pH values of the *Spirulina* sp. protein extract and *Isochrysis* sp. protein extract were determined to be 4.25 ± 0.00 and 4.29 ± 0.05, respectively. Both of these physicochemical characteristics are important for the storage of any food product before further use in industry. Foods with a high $a_w$ value show rapid deterioration due to biological and chemical changes, due to the fact that $a_w$ influences microbial growth, lipid oxidation and enzymatic activities in foods [26,27]. The pH values of both protein extracts are comparable to those of protein extracts reported previously from the brown macroalga *Himanthalia elongata* [9], which has a pH value of 3.99 ± 0.02. The water activities identified for the *Spirulina* sp. and *Isochrysis* sp. protein extracts were similar to those of sliced almonds, which have a reported $a_w$ of 0.476 at 20 °C [19].

### 3.4. Foaming Capacity and Stability of Protein Extracts

The foaming capacity and stability of proteins can have a sensory and flavor impact on food formulations and can improve the smoothness, lightness and palatability of the food product [28]. The foaming capacity and stability profiles of the protein extracts generated from *Spirulina* sp. and *Isochrysis* sp. were compared at different pH values to the foaming capacity and stability of flaxseed and whey protein isolates in triplicate (Figure 1). The foams formed for the microalgal protein extracts were most stable at pH 2 and pH 4. Otherwise, the pH level had little effect on the foaming stability.

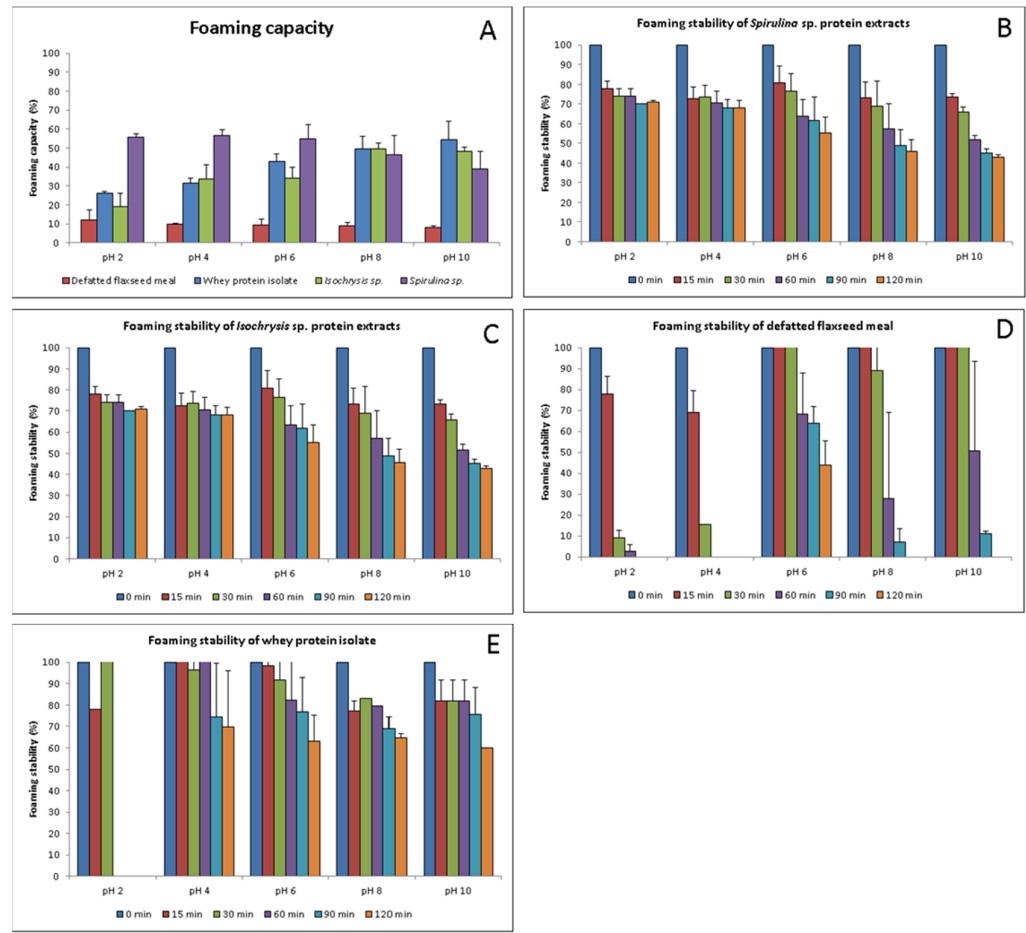

**Figure 1.** Foaming capacity (A), as well as foaming stability of *Spirulina* sp. (B), *Isochrysis sp*. (C), flaxseed meal (D) and whey protein isolate (E) (n = 3).

### 3.5. The WHC and OHC of Protein Extracts

The hydration characteristics of food are very important for effective digestion to occur and can be defined by the swelling capacity, solubility, WHC and other characteristics. The WHC of a food product describes the ability to hold water within the food matrix [29]. Furthermore, the WHC of foods such as meat relates to quality attributes, which influence product yield and have economic implications but is also important in terms of eating quality [30]. The WHC and OHC of the protein extracts generated from *Spirulina* sp. and *Isochrysis* sp. were compared to those of a whey protein isolate and flaxseed protein, respectively (Figure 2). The WHC values for the protein extracts generated from *Spirulina* sp. and *Isochrysis* sp. were 2.25 g water/g protein extract and 0.47 g water/g protein extract, respectively, compared with a WHC of 6.39 g water/g defatted flaxseed protein. Previously, De Moor and Huyghebaert [30] reported that the overall water-holding capacities of whey powders and demineralized whey powders are generally low, and this is what we observed in this study. These WHC values were lower than those reported previously for seaweed-derived protein extracts, including *Himanthalia elongata* (WHC value of 10.27 ± 0.09 g water/ g protein) [9]. The WHC obtained for the *Spirulina* sp. protein extract (2.25 g water/g protein) was similar to that derived from *Kappaphycus alvarezii*, which had a reported WHC of 2.22 ± 0.04 g water/ g protein [31]. High WHC values are required for viscous foods including custards, sausages, doughs and baked products, as they help to provide thickening and viscosity to the foods [32]. The WHC values observed for both algal protein extracts were lower in WHC value than the *H. elongata* proteins [9]. The solubility values observed for both protein extracts at pH 4 and 2, respectively, might be

because the proteins have their isoelectric point at these pH values, and at more acidic or alkaline pH values, the protein will acquire a net positive or negative charge.

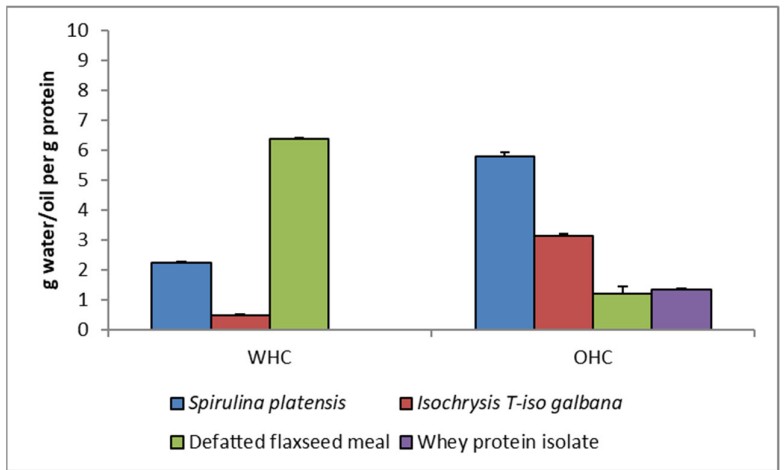

**Figure 2.** Oil and water holding capacity of microalgal protein extracts compared with the controls (n = 3).

The microalgal protein extracts had greater OHCs than the whey protein isolate and flaxseed protein analyzed. A high OHC value is desirable in food manufacturing for the retention of flavor and improved palatability. The OHC of the *Spirulina* sp. protein extract was 5.80 g oil/g protein extract, and the OHC for the *Isochrysis* sp. protein extract was 3.16 g oil/g protein extract, compared with 1.23 g oil/g defatted flaxseed protein extract and ~1 g oil/g whey protein isolate. These OHC values compare favorably to other protein isolates, including the seaweed *H. elongata* protein concentrate, which reported an OHC of 8.1 g oil/g protein extract [9]. The macroalgal protein powder *Kapparazii powder* ™ had an OHC value of 0.20 g/g, and rice bran protein concentrates have reported OHC values of between 3.74 and 9.18 g oil/g [29].

*3.6. Molecular Weight Distribution Profile of Soluble Proteins Only from Spirulina sp. and Isochrysis sp.*

The molecular weight distribution profiles of the soluble protein extracts generated from *Spirulina* sp. and *Isochrysis* sp. are shown in Figures 3 A,B. The profiles varied between the two microalgae, with the largest percentage of proteins in the *Spirulina* sp. extract found in the >100 kDa fraction (62.28%). However, this fraction, in fact, not only contained >100 kDa proteins, as seen by the appearance and distribution (Gel A). A lot of the content was found in the lower part of the gel, indicating a MW far less than 100 kDa. It also contained protein >100 kDa. Based on the SDS-PAGE of the individual fraction, the protein in the individual fractions did not have the overall characteristics (i.e., sharp size distribution) intended. Furthermore, when considering the whole extract, the content of high MW protein (>100 kDa) did not appear to be as high as what was determined by the fraction of dry mass. This calls into question the efficiency of the fractionation and that it may not have worked as intended. Size exclusion chromatography could also be used to determine the size range of the extracts. This is in contrast to the protein extract generated for *Isochrysis* sp., where 70.54% of the proteins were low molecular weight proteins. The large percentage of proteins that were of a low molecular weight in size (less than 3 kDa) in the *Isochrysis* sp. protein extract was positive in terms of the functional food applications of this protein, as it is well known that peptides and proteins less than 3 kDa in size can be bioactive in nature and may have health-beneficial effects for the consumer [33, 34]. However, this might also indicate that the traditional extraction method is not suitable for



use with *Isochrysis* sp., as less than 10% of the total amino acids found in the dry microalgae were extracted and found in the soluble protein extract. Similar results were shown in a recent study where traditional protein aqueous extraction was applied to the seaweed *Eucheuna denticulatum* [35].

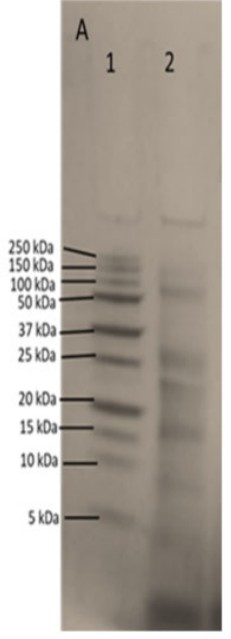
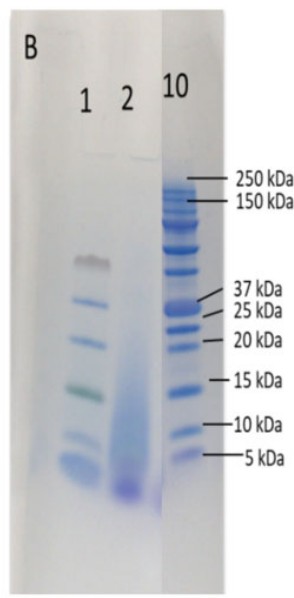

**Lane 1**; High MW marker, **Lane 2**; Whole *Spirulina* sp., traditional extract protein,  **Lane 1**; Molecular weight marker; **Lane 2**: *Isochrysis* protein traditional extract;

**Figure 3.** Tris-Tricine SDS-PAGE, performed using a Mini-PROTEAN® electrophoresis unit using Mini-PROTEAN® 10−20% Tris-Tricine precast gels (Bio-Rad laboratories, USA) for *Spirulina* sp. protein extract (A) and *Isochrysis* sp. protein extract (B).

### 3.7. Solubility of Protein Extracts

The solubilities of the microalgal protein extracts were assessed at pH values varying from 2 to 12 at concentrations of 1% *w/v* in water (Figure 4). The pH had a significant influence on the solubilities of both microalgal protein extracts. The minimum solubility was observed at pH 4 (4.99%), rising to 32.44% at pH 10, with the maximum solubility of the *Spirulina* sp. protein extract observed at pH 12 (62.99%). The minimum solubility value observed for the *Isochrysis* sp. protein extract was at pH 2 (14.12%), with a maximum solubility at 1% *w/v* observed at pH 12 (19.25%). The maximum solubility values for both whey and defatted flaxseed protein were also observed at pH 12, with solubility values of 100% and 87%, respectively, observed for these proteins when tested at pH 12 at a concentration of 1% *w/v* water.

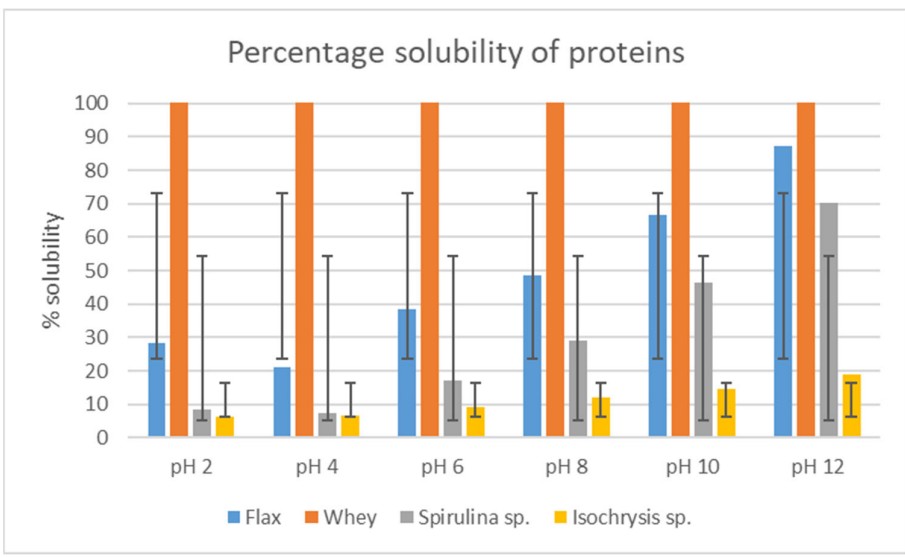

**Figure 4.** Solubility of extracted proteins at different pH values (n = 3).

*3.8. Emulsion Activity and Stability of Protein Extracts*

The emulsion activity and stability of the protein extracts generated from *Spirulina* sp. and *Isochrysis* sp. at five different pH values using four different oils are shown in Figure 6. The *Spirulina* sp. protein extract suspended in olive oil showed the highest emulsion activity (22.41%), and rapeseed oil showed the lowest (21.00%). The *Isochrysis* sp. protein extract suspended in olive oil showed the highest emulsion activity (11.36%), and rapeseed oil showed the lowest (10.01%). Both microalgal protein extracts displayed poor emulsion activity and stability percentage values compared with the commercial controls of defatted flaxseed protein and whey protein isolate (Figure 5). However, the *Spirulina* sp. protein extract did display excellent emulsion stability values (85.91%) when assessed in olive oil. The EA values obtained here are in contrast to results reporting the EA values of seaweed derived protein extracts in oils (75.68–99.67%). In general, emulsion stability was more effective in the olive and groundnut oils. Emulsion stability can be affected by several factors, including the pH, droplet size, net charge, interfacial tension, viscosity and protein conformation. The high emulsion stability observed previously for the *Spirulina* sp. protein extract in olive oil after heating at 85 °C could be due to the dissociation of some proteins, resulting in the formation of subunits with more hydrophobic groups and stronger interactions with the lipid phase [36].

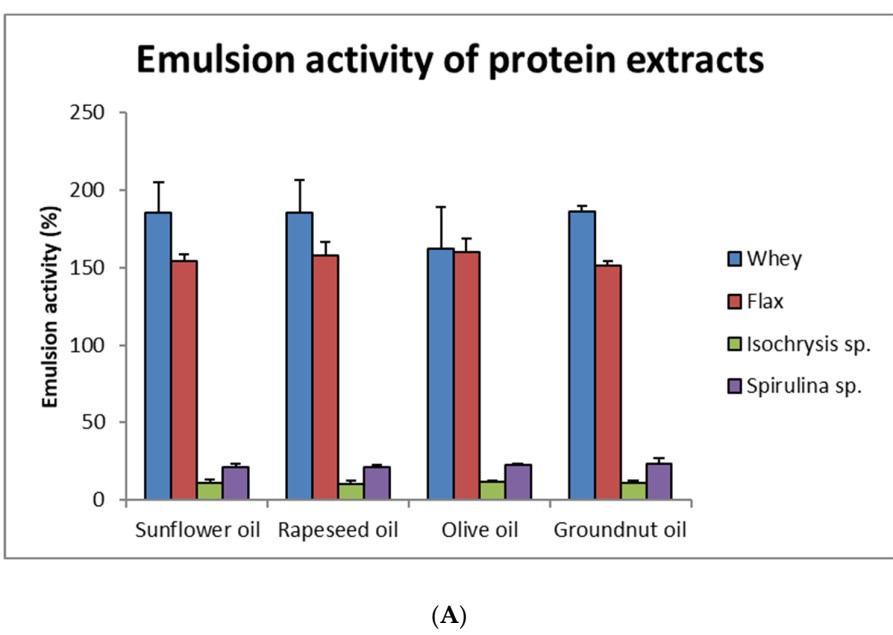

(**A**)

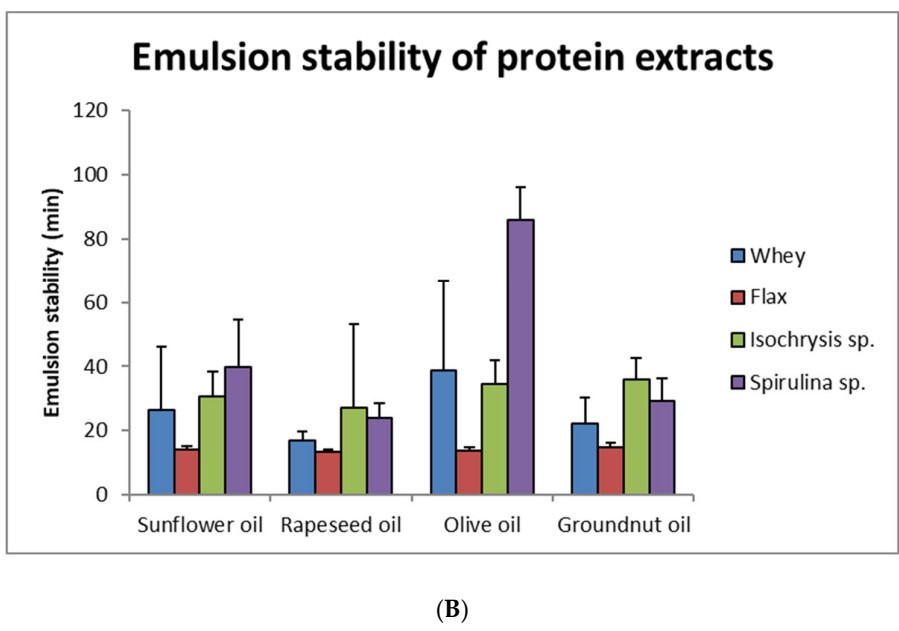

(**B**)

**Figure 5.** ES (**A**) and EA (**B**) of soluble protein extracts (n = 3). Values calculated against whey.

### 3.9. ACE−1 and Renin Inhibitory Activities

The microalgal protein extracts displayed renin inhibition values ranging between 23.18% (± 2.16%) and 25% (± 7.7%) (Figure 6) for the *Isochrysis* sp. and *Spirulina* sp. protein extracts, respectively. The *Isochryis* sp. protein extract inhibited ACE−1 by 95.34% compared with the *Spirulina* protein extract, which inhibited ACE−1 by 91.04% when assayed at a concentration of 1 mg/mL compared with the control Captopril®. There was significantly lower renin inhibition activity compared with ACE−1 inhibitory activity. The characteristics of renin inhibitory peptides are not as well defined as other bioactive peptide targets (such as ACE-I inhibitory peptides) [25,36].

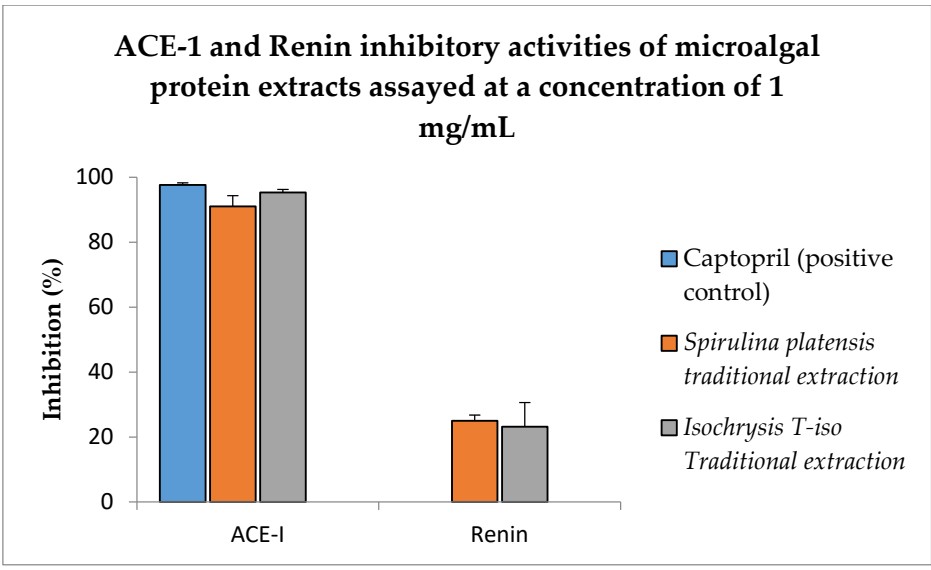

**Figure 6.** ACE−1 and renin inhibitory activities of proteins extracted from *Spirulina* sp. and *Isochrysis galbana T-Iso*, assayed at a concentration of 1 mg/mL compared with the control (Captopril©) (n = 3) and 10 μM Z-Arg-Arg-Pro-Phe-His-Sta-Ile-His-Lys-(Boc)-OMe (renin inhibitor).

## 4. Conclusions

The foaming capacity and stability values observed for both microalgal protein extracts, along with their solubility at pH 10–12 and excellent OHC values, suggest that these proteins could be used in the formulation of a wide variety of foods with alkaline pH values and with an oil consistency such as that in green drinks with a pH greater than 9.5 and tomato juices. Based on the FC and FS, the microalgal extracts have good stability independent of the pH value compared with flaxseed. However, the aqueous solubility of the extracts limits applications. The renin-angiotensin-aldosterone system (RAAS) controls hypertension development through the actions of two enzymes: renin (EC 3.4.23.15) and ACE−1 (EC 3.4.15.1). In a previous study, the response surface methodology (RSM) was used to generate hydrolysates with ACE−1 inhibitory activities from *Isochrysis galbana* (not *Spirulina* sp.). The conditions that produced the greatest ACE−1 inhibitory activity were a temperature of 55.64 °C, a substrate concentration of 5.46 g (100 mL)$^{-1}$ and a trypsin enzyme/substrate ratio (E/S) of 6.27% [37]. The highest ACE inhibitory activity observed was 47.62% ACE−1 inhibition at a concentration of 1 mg/mL [33]. The ACE−1 inhibitory values observed in this study (95%) were greater than 47.62% at a concentration of 1 mg/mL compared with Captopril©, the positive control which was assayed at a concentration of 0.05 mg/mL. The renin inhibitory control was also assayed at the same concentration. In addition, the protein extracts generated were found to inhibit renin [38]. In addition, the observed ACE−1 and renin inhibitory activities shown by the protein extracts suggest that they could find applications in functional foods for preventative healthcare. *Spirulina* is certified as generally recognized as safe (GRAS)—GRN No. 127—by the United States (US) Food and Drug Administration (FDA) and complies with the requirements for use set out by Regulation (EU) 2015/2283 on novel foods [25]. However, the presence of contaminants and anti-nutritional factors as well as allergens should be evaluated as discussed previously. The bioaccessibility and bioavailability of the ACE−1 inhibitory activities should be characterized, especially when these ingredients are incorporated in different food matrices, a driving factor of such important nutritional and health benefits [38]. This data is essential for preventative or adjuvant therapeutic strategies against high blood pressure and associated diseases.

**Author Contributions:** Conceptualization, S.B. and M.H.; methodology, S. B.; writing—original draft preparation, S.B.; writing—review and editing, M.H.; supervision, M.H.; project administration, M.H.; funding acquisition, M.H. All authors have read and agreed to the published version of the manuscript.

**Funding:** Teagasc grant number NFNY–6889–142 funded this research.

**Acknowledgments:** This project was carried out as part of the BioAlgae Project (Teagasc funded project no. 2016–073). Stephen Bleakley is in receipt of a PhD Walsh Fellowship (grant no. NFNY 6889–142) funded by Teagasc for the period of 2016–2020. The authors would also like to acknowledge the EUALGAE COST Action – European Network for algal bioproducts ES1408.

**Conflicts of Interest:** The authors declare no conflict of interest.

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
