# Peer review of "Functional and Bioactive Properties of Protein Extracts Generated from Spirulina platensis and Isochrysis galbana T-Iso"

_applsci, doi:10.3390/app11093964_

Round 1

Reviewer 1 Report

The manuscripts investigates the functional and bioactive properties of protein extracts from two microalgal species using a range of different functional assays, and thus adheres very well to the topics outlined for this special issue of Applied Sciences.

Overall the manuscript is quite well written in terms of language and understandability with only minor proposed revision. However, there are quite a lot of both major and minor aspects related to the content, which should be addressed and accounted for in my opinion. Most aspects are very specific, and only a few general comments. In the following, some of the aspects will be outlined and some suggestions for improvement provided. The vast majority should be fairly easy and fast to address. However, some data are missing (SDS-PAGE and EAI/ESI), which should makes overall assessment difficult. This should be addressed.

The authors have chosen to have results and discussion separately, but it seems that they are really somewhat interconnected in their use in the current form. The authors should consider either combining the two sections into one or move a significant amount of the text from results to discussion. This is both as several paragraphs in the results section seems more like discussion/evaluation, and that the discussion is very brief and vague in its current form. If the authors wish to retain the current structure, the results section should only include a fully objective presentation of the data and not discuss nor evaluate them. And more elaboration/discussion/evaluation in the discussion is advised.

Standard deviations are missing in several figure (1, 2, maybe some of the bars in figure 3?, 4, 5, 6, maybe 7 but this figure is missing). Please include if available and see the comment below about replicates.

The authors should re-check references, as there appears to have been a shift at some point. The refs indicated in the text e.g. 16 and 17, are likely to be 17 and 18 in the reference list.

The study seems thorough and quite well designed, however, the presentation of the data and the discussion hereof does not do the data justice in my opinion. The authors should to a higher degree highlight what is excellent in these extracts. Moreover, the authors should also try to be more clear on what the novelty in their study consists of, as it, to a large degree, takes on the form of a technical report more than a novel research project. By doing so and thoroughly addressing the suggested revisions below, I do think that the data presented and the content of the study would be of both interest and relevance to the readers of this special issue.  

Major aspects:

Abstract: The abstract is mostly a summary of the methods applied and they properties determined in general terms (i.e. a lot of “was determined/assed” an very little information of what was actually found). The only thing related to the actual finding is the last sentence, which vaguely suggests a potential for the application in foods. This leaves the reader uninformed based on the abstract. I suggest a revision (especially L.17-24) where the authors to a much larger extent focus on what the study actually showed. I would like to stress that I am not suggesting overloading the abstract with specific data, but rather that the extracts are evaluated in terms of the determined properties – especially since the authors provide data for comparison to commercial whey protein and defatted flaxseed protein.

L.44-56: The authors should consider to revise this section to better reflect aspects that are more directly related to their own study. This could for instance be concerning the same species as investigated here or the same extraction method applied on other microalgal species. The section seems a bit unstructured and the objective is unclear in relation to this study. The introduction does not seem to cover state of the art sufficiently. The authors could, for instance, consider the works listed below or even their own 2017 review in Foods as further references/inspiration.

Bertsch, Pascal, et al. "Proteins from microalgae for the stabilization of fluid interfaces, emulsions, and foams." Trends in Food Science & Technology 108 (2021): 326-342.

Koyande, Apurav Krishna, et al. "Microalgae: A potential alternative to health supplementation for humans." Food Science and Human Wellness 8.1 (2019): 16-24.

Gerde, Jose A., et al. "Evaluation of microalgae cell disruption by ultrasonic treatment." Bioresource technology 125 (2012): 175-181.

Sankaran, Revathy, et al. "Extraction of proteins from microalgae using integrated method of sugaring-out assisted liquid biphasic flotation (LBF) and ultrasound." Ultrasonics sonochemistry 48 (2018): 231-239.

L.101-102: I suggest rephrasing here, as the stated expression would not determine the protein content of the extracts. The protein content of the extract would be relative to the dry weight of the extract and not the dry biomass. This is more the yield of extracted protein relative to dry biomass. Particularly as protein content (relative to extract dry weight) is also reported (L.242)

L.110: As the amino acid composition was not known at this point, how were supernatant diluted to an approximate molarity of each individual amino acid? And was a dilution made for each amino acid in order to have approximately 250 nmol of the respective AA (i.e. up to different 20 dilutions for each sample)? This seems unclear

L.121: Please check denaturing temperature. 55°C is far below recommendations (usually 90 or 95°C)

L.171: I do not agree with the definition of FC as being VF minus V0. Rather it should be relative (i.e. VF divided by V0). See for instance the references below.

Bao, Z.J.; Zhao, Y.;Wang, X.Y.; Chi, Y.J. Effects of degree of hydrolysis (DH) on the functional properties of egg yolk hydrolysate with alcalase. J. Food Sci. Technol. 2017, 54, 669–678.

Jafarpour, Ali, et al. "Biofunctionality of enzymatically derived peptides from codfish (Gadus morhua) frame: Bulk in vitro properties, quantitative proteomics, and bioinformatic prediction." Marine drugs 18.12 (2020): 599.

Section 2.11: This approach does only give the distribution for the soluble proteins and not the extract as a whole. This should be clearly and explicitly stated – especially based on the somewhat modest solubility of the extracts. The authors should also reflect upon the fact that MWCO spin filters are not clear MW cutoffs, but based on pore sizes, which again depend on the folding state of the individual protein. As such, it can be dangerous to over interpret the results of such a fractionation, although is does provide some estimate. A more reliable approach could be to do e.g. size-exclusion chromatography or even quantitative analysis of the SDS-PAGE analysis. The lyophilized extracts could also be directly solubilized and boiled in sample buffer which would very likely increase the overall protein solubility. This would be a goof comparison to running resolubilized protein on SDS-PAGE and would also give insight into which proteins in the extracts are insoluble.

Where is the SDS-PAGE analysis described? I cannot seem to find the data/image.

L.220: Please rewrite the expression in a more meaningful way. This is very hard to decode. The same goes for L. 230. Consider writing the expression with symbols and explain the symbols in the text below.

L.221-230: Was no control used for Renin inhibition?

L. 263: The low protein content/total amino acid content of the isochrysis extract should be discussed, as this is less than a factor 10 of the content of the whole biomass. Please include in the discussion.

Figure 2B: As the free amino acid content is determined, this should be presented in the text and used actively. Otherwise, these data are redundant and should be excluded or moved to the SI.

L.289-291: Is this statistically significant? And what does the authors mean by “levels”? For instance, the FC seems higher for isochrysis at higher pH. Also, this statement does not seem to be in agreement with what is concluded (L.418-421)

L.303: Are they really comparable? Were replicates performed and are the differences statistically significant? Also, does the Whey have zero WHC as presented in Figure 4? If so, then the microalgal extracts would be better and not comparable to whey.

Section 3.6: I am missing a stance by the author of the MW distribution (presented in a discussion nonetheless). Why do the authors think there is such a tremendous difference? Were there any differences in treatment in your lab or at the producer/supplier? It seems very non-representative of a natural distribution to have such a distribution as found in isochrysis. Could this be a result of partial hydrolysis? Or could this be linked to the low extraction yield (less than 10% of the total amino acids in the extract compared to the raw biomass)? Perhaps the process is not suitable for this organism, which has previously been shown for e.g. seaweed proteins (E. denticulatum) by aqueous extraction (see preprint listed below).

Gregersen, Simon, et al. "Proteomic characterization of pilot scale hot-water extracts from the industrial carrageenan red seaweed Eucheuma denticulatum." bioRxiv (2020).

Figure 5: Please revise number of significant figures in this figure. Also, what are “other”? In addition, the missing SDS-PAGE image could be very informative to have included as a “B” part in this figure for comparison. This comparison should also be addressed actively by the authors in the discussion, where also the influence of pH on solubility could be included in a grander assessment. For instance, was the pH of the protein solution for both the fractionation and the SDS-PAGE the same? Do the authors think this would change anything if done at other pH etc.?

Section 3.8: These data appear to be missing! Figure 7 does not show ESI/EAI but inhibition. Please fix this.

Section 3.8: The listed emulsion activities are listed in %, but the presented method (section 2.10) gives the EAI in m^2/g. This should be addressed. Or is this relative to something?  The missing figure makes it very difficult to asses.

L. 365-367: Is it a reasonable comparison to compare microalgal protein extracts to seaweed protein extracts 1:1 in such a way? Also, the state of the proteins (i.e. native or denatured) would be very important and should be addressed. This would also require a stance by the authors on the impact of the extraction method, which seems to be more general topic that could have a separate section in the discussion.

L.370-373: Where is this heating step described? And does this only apply to spirulina? What would the absence of heating mean in the opinion of the authors? Was this assessed? The missing figure makes it hard to evaluate.

L. 375: Was no control used for Renin inhibition?

L.377-379: This paragraph reads very difficultly. Is the inhibition relative to the control? Or relative to initial activity as outlined in L. 220? And if so, what was the inhibition of the control inexact numbers? The use of “compared” seems faulty in this paragraph. Were differences statistically significant?

L.391-392 + L. 421-423: That is probably good, but so what? Does that mean that they are good? What about the rest of the AAs? Such short discussion does not support the conclusion that they have excellent AA profiles (L. 421-423). And excellent in comparison to what? That all essential AAs are present cannot be a surprise as this is a full extract – rather the levels are of importance here.

The previous comment also applies to the remaining part of the discussion (L.391-407). The results should be elaborated, evaluated, and discussed to a much higher degree. In the current form, the discussion is merely listing value from other studies without stance or discussing implication. It is very similar to a lot of sections found in the results (which I suggest moving if not combining results and discussion).

L. 397-400: But how does it compare to whey?

L. 404-407: Please revise and be more explicit. “The solubility values observed” requires the reader to go back and to make their own evaluation. Be specific. If this is because the lowest solubility was observed here, then state that. And following this, a more acidic pH than 2 is very hard to imagine being relevant in foods (L.405). Please revise this part.

L. 415-416: So this means that unhydrolysed protein is better than tryptic hydrolysates? This should be highlighted very clearly and “celebrated” as this appears to be a major advantage of the obtained extract without the need for further processing. Also “renin inhibitory (activity) was also observed [31]”. So what was the renin inhibitory in that study? And how does it compare the your extract? Or is this statement related to your own data? If so, this paragraph should be revised for clarity.

L.423-431: This part does not seem suited for a conclusion but rather a discussion. References in a conclusion is IMO rarely a good idea. Rather, the authors should reflect more on the perspectives of their data rather than the limitations here. For instance highlight that the high ACE-I inhibitory activity of the crude extract appears to be so good and even better than hydrolysates of the same species (L.415), and as such, the protein extracts are promising not only as functional ingredients but also have bioactivities that could promote health. In this regard, the authors may also want to consider the potential presence of non-protein bioactive compounds in the extract – this could also be a topic in the discussion.

Minor aspects:

L.34: please elaborate on “also rich” since you present specified protein content for other sources above. Please also indicate if this is drymatter basis or crude, as this makes comparison easier for the reader.

L.100: Please indicate a suitable reference for using a N-to-protein conversion factor of 4.4. This is listed in “results” (L. 239) as [16], but I think it would be better suited here. Also, please check if it is not in fact [17], that is the correct reference for the factor.

L.244-246: Please insert reference again although cited earlier. Makes it easier for the reader to identify the specific reference claiming this without having to go back in the manuscript. Moreover, the authors should consider to cite the original research stating this protein content in platensis rather than a review.

L.106-109: What is the reason for TCA precipitation at this stage? Should the proteins not be fully acid hydrolysed?

L.119: If protein extracts were diluted 1:1 with loading buffer, what was the initial concentration? Because, the extracts were earlier lyophilized.

L.130: The pH of an aq. resuspension is not really a profile but rather single data point.

Section 2.8: In stead of determining the relative solubility at a given concentration (10 g/L) as a function of pH, it would be more informative to quickly calculate the absolute solubility at that pH (given that the solubility of the whey protein would be expressed as >10 g/L, as this was fully soluble at 1% based on the data)

L.197-199: Would the authors expect a pH dependence in the size distribution it assayed at different pH? This could be elaborated on later in the manuscript.

L.223-225: Please rephrase. What does “each sample inhibitor at a concentration of 1 mg/mL DMSO” mean? Was the protein extracts at 1 mg/mL in neat DMSO or was the DMSO at 1 mg/mL (as it it reads now)?

L.231: Does the triplicate determination refer to all experiments or only the ACE/Renin assays? Replications should be mentioned explicitly for each experiment.

L.244-246: This part seems more suitable in a discussion, and should the influence of the extraction method should be emphasized. Also the non-protein nitrogen should be discussed.

The authors could consider gathering Figure 1 and 2 in a table in stead – seems a bit much to use two Figures for this, and the exact numbers would be nice to have in a table format. The figures could be moved to SI.

L.260-262: Please remember italics for all mentions of the microalgae

L.259-267: Please revise this section – it is not easy to read and the species are mixed in the sentences. In particular L.260-265 should be broken up into shorter sentences.

L.267: This feels more like evaluation/discussion of the results and should be moved.

Section 3.3 (L. 275-279) These data could be included in the same table as Figure 1+2 for better clarity.

L.279-283: This feels more like evaluation/discussion of the results and should be moved.

L.297-302: This part seems to be better suited in a discussion/evaluation of the data, its implications and applicability.

L.306-312: This feels more like evaluation/discussion of the results and should be moved.

L.317-318 + 321-325: This feels more like evaluation/discussion of the results and should be moved.

L.320-321: A very specific value with four significant figure (are so many really necessary in general? I would feel three is enough I all of section 3.5) are given for the microalgal extracts and the flaxseed, while ~1 is given for whey. Please be consistent in use of significant figures.

L.333-336: This feels more like evaluation/discussion of the results and should be moved.

L.347-350: From Figure 6, the solubility of Whey appears to be 100% regardless of pH?

L.365-373: This feels more like evaluation/discussion of the results and should be moved.

L.368-369: Please indicate suitable reference(s).

L.380-383: This feels more like evaluation/discussion of the results and should be moved.

L.400-403: Please revise sentence for clarity.

L.408-409: Please indicate suitable reference

L.409-411: please indicate the reference for this study here.

L.418-421: Based on the FC and FS, the microalgal extracts may be applicable in an even broader range, as they show, largely, good stability independent of pH. Especially compared to flaxseed. However, taking aq. solubility into account, I do agree.

Author Response

Dear reviewer,

Many thanks for your detailed review of this paper. Please find attached our response to your suggested edits -highlighted in green in the attached word document. We hope we have addressed most of your queries. We have included SDS PAGE and EAI/ESI data. We have included suggested references and other items mentioned.

Thank you and Kind regards,

Maria

Reviewer 2 Report

The paper deals with the techno-functional and bioactive properties of proteins from micro-algae. The treated topic is current and the results expand knowledge about this novel source of plant proteins.

However, the text has some shortcomings or inaccuracies:

  • In the paragraph 2.4 (materials and methods) Authors describe the SDS-PAGE, but no gel is shown in the results, although it would be interesting to see the electrophoretic pattern.
  • Lines 231-234 should be inserted in a new paragraph 2.13 “Statistical analysis”.
  • Lines 244-246: please, cite the studies.
  • Authors should check the reference numbering, in particular from citation [18] to [30]. For example, I think that citation [18, 19], in line 283, are [19] and [20] in the references list, and so on…
  • The WHC value for whey protein is not reported, neither in the text and in figure 4. Why?
  • Figure 7 (about emulsion activity) is missing and Figure 7 (about ACE-1 and renin inhibition activity) should be Figure 8.
  • In some figures the standard deviation of histograms in not reported (figures 1, 4, 6).
  • I suggest to merge the paragraphs Result and Discussion in a one chapter (Results and discussion).

Author Response

We thank the reviewer for their detailed review of our paper. We have addressed the queries raised in the attached document.

Kind regards,

Maria

Reviewer 3 Report

It has been a great honor, as well as a pleasantly challenging activity, to review the article entitled "Functional and bioactive properties of protein extracts generated from Spirulina platensis and Isochrysis galbana T-Iso" by Stephen Bleakley and Maria Haye. The article is structured following the classic model for this type of material (Article), comprising five parts: Introduction, Materials and Methods, Results, Discussion, and Conclusions. This study investigated functional and biological activity of Spirulina platensis and Isochrysis galbana T-Iso-derived protein. This information is very useful not only for the food industry in selected beverages but also for cosmetics industry. In my opinion the introduction of the manuscript is very good, gives all the essential information for the understanding and analysis of the work developed. I recommend to the authors add the statistical analysis in Figure 3.

Author Response

Dear reviewer,

Many thanks for your kind review of this paper. We have added statistical work on all Figures in the revised manuscript.

Kind regards,

Maria

Round 2

Reviewer 1 Report

Please see attached word document.

Author Response

Dear editor and reviewer - Please find attached our response to reviewer comments - 12 pages - Kind regards, Maria

Reviewer 2 Report

The Authors included reviewer's suggestions in the revised manuscript, improving it. 

Author Response

Thank you for the positive review of our paper - Kind regards, Maria

Round 3

Reviewer 1 Report

I would initially use the opportunity to send my deepest appreciation to the authors for their very positive and detailed way of dealing with my comments and suggestions. And for elaborating on points where we are not in full agreement. This is, in my opinion, very constructive and has elevated the level of the manuscript overall.

The absolute majority of points and comments have been addressed in a highly satisfactory manner in the latest revision. Unless otherwise mentioned bellowed, I consider all issue sorted. Only remaining issues will be outlined below.

L9-10: To elaborate on my previous comment, in the first revision, the term “methods” (plural) was used. That was the basis for the comment on multiple methods. IMO the added “one” could easily be replaced by e.g. “a” if the authors prefer. But this may just be semantics. And thank you for adding the info on the method here as well. It was exactly the level of detail I was thinking at this point.

L176: The equations look much better now, but please make sure that the subscript is applied in the equation, and not just in the caption below.

L192: The subscript for “c” in Fc is missing in the caption for the equation.

L241-242: Please remember the subscripts in the caption for the equation

Section 3.2: I highly appreciate the revision, but struggle to understand how FAA can be larger than TAA. Are free AAs not included in the determination of the total content? If not, then I would argue that it is not total AA, as this parameter by definition should cover both bound and free AAs. From M&M (Section 2.3) it is also a little hard to decode how the determination of the two parameters differ, as, IMO, it reads as one protocol description (and based on the description, it includes acid hydrolysis and would thus determine TAA). Where do the determinations of FAA and TAA differ? The authors should consider to make this clear in M&M. And consider if it is in fact total AA, they determine, and if yes, how the total content can be lower that the free content. It is counterintuitive to me. Finally, I also struggle to understand the large inconsistency between determined protein content (by %) and the TAA content for the two extracts. Both contain >70% protein (i.e. >700 g/kg) but the total AA content is max 7% (i.e. 70 g/kg). So for Spirulina, the discrepancy is a factor 12 while for Isochrysis, it is a factor 108. Why is this? I fully get that the method used is crucial (as discussed around L264), but a difference of more than 100-fold cannot just be based on the method IMO. That the extracts are somewhat depleted for free AA also makes sense based on the treatments used during extraction, but the protein content, TAA, (and FAA for that matter) within the same sample, should correlate in a reasonable manner. I would urge the authors the double and triple check their data to make absolutely sure there are no unfortunate errors. If that is not the case, a brief comment on inconsistencies would be suitable.

Section 3.6: I would like to thank the author for addressing this section and taking a constructive dialogue even though we may disagree. Based on the reply from the authors, I may not have been clear in my previous report, for which I apologize. I am not questioning that the SDS-PAGE analysis shows what it does – that is unquestionable. I also fully acknowledge that the >100kDa fraction does appear to contain the most (total) protein (by mass). My concern was more that this fraction in fact does not only contain >100 kDa proteins as seen by the appearance/distribution of gel1-lane8. A lot of the content is found in the lower part of the gel indicating a MW far less than 100 kDa. It does also contain protein >100 kDa (as the only fraction, which is of course great and as intended), but it is all the other things in there, that is my concern (as agreed by the authors). This was my original point and why I questioned the efficiency of the ultracentrifugation approach employed for fractionation. The proteins are fractioned – without any doubt – but based on SDS-PAGE of the individual fraction, the protein in the individual fractions to not have the overall characteristics (i.e. sharp size distribution) intended. Furthermore, when considering the whole extract (gel1-lane2), the content of high MW protein (>100kDa) does not appear to be as high as what was determined by fraction dry mass. The again makes me question the efficiency of the fractionation and that it may not have worked as intended. I hope my point is more clear now. I am not questioning if the authors have done the experiment properly, but only that the outcome of this approach is not as clear cut as one would like it to be. And thus, care should be take when interpreting the data. I do fully agree that gel2-lane2 (isochrysis extract) appears to be mostly low MW protein based on SDS-PAGE as well. Overall, I think the authors should feel free to include the SDS-PAGE data for the fractions as well, as it is indeed highly informative and illustrative (and don’t feel forced to crop images as seen in the manuscript now – although otherwise indicated earlier in the response?). But the manuscript should, in my opinion, also include reservations on the fractionation in the text based the appearance, unless the authors completely disagree with the point. An alternative possibility could be to giver the fractions consecutive numbers, thereby not stating explicitly, that the >100kDa fraction only contains >100kDa protein, which it does not seem to do. This way, the authors can, in my opinion, easily include the fractions, and just briefly explain that it is not a clear cut-off fractionation, but that that e.g. the fraction now called >100 kDa (which would be fraction 6) is enriched for proteins >100kDa and that e.g. fraction 2 is enriched for protein in the range 3-10 kDa etc. This would also allow the authors to include the now removed data, which I may have been too harsh in my judgement of. After very careful and long consideration, I may have been too blunt in my earlier statement, for which I apologize. I still stand by my concerns and comments, but also acknowledge that this may have been addressed differently. For any inconvenience this has caused, I do apologize. The issue can be addressed in a satisfactory manner by including some reservations and comments in the text and by alternative nomenclature.  We do not have to agree, but I would just like the authors to consider my point, and feel free to include the data, but mention the limitations/reservations explicitly in the manuscript text. Because as it is now, section 3.6 includes data on relative protein content not described in M&M (as the fractionation protocol has been removed) nor presented in the manuscript. If kept in the current form and the authors decide not to include the now omitted data (with the reservations/limitations requested), the section is undocumented. The authors may decide the solution they see fit for this matter, but a final revision is needed.

Figure 5: I would urge the authors to check this figure. In particular, the error bars do not match with the bars in the histogram (e.g. flax at pH4, the error bars are above the upper limit of the histogram while for e.g. flax, spirulina, and isochrysis at pH12, the error bars do not reach the upper limit of the histogram bar). Whey has no error bars at any pH. Also, based on the text (L400-403), the solubility of flax at pH 12 was 100%, but the figure shows ~87% (as indicated in the two former versions of the manuscript). Please check. Furthermore, I also previously asked it whey was fully soluble at all pH values tested (the histogram shows 100% at all pHs). If this is the case, then that is absolutely fine, I just want to make sure that it is indeed the case. Lastly, and just a minor thing, I would suggest to use the same software/template to plot all data, as Figure 5 now looks different from the rest.

Section 3.8: I still do not understand why in the text (and figure 6A), emulsion activity is given in % when the used expression would provide a number with the unit m2/g? If the unit (%) is correct, this would indicate that the data has somehow been normalized to a reference value (100%). If this is the case, please describe the reference point and the procedure briefly in M&M.

L.429-432: I previously commented “At what point is the emulsion heated to 85°C? This is not described in M&M.”. I do not feel this is addressed. If this paragraph is intended as a potential explanation of why the olive oil emulsion is more stable, and this is justified by others having observed stability after heating a spirulina/olive oil emulsion to 85°c (ref 36), please revise the sentence for clarity.

Conclusion: I suggest moving L466-476 to section 3.9, as this seems more like discussion than conclusion. L466-468 would be a good introduction to section 3.9, while L468-476, would be a good ending to the section, putting the obtained data in perspective.

Author Response

We have edited the paper and addressed the comments of reviewer 1 as requested.

Kind regards,

Maria
